# The Influence of Food Regimes on Oxidative Stress: A Permutation-Based Approach Using the NPC Test

**DOI:** 10.3390/healthcare11162263

**Published:** 2023-08-11

**Authors:** Agata Zirilli, Rosaria Maddalena Ruggeri, Maria Cristina Barbalace, Silvana Hrelia, Luca Giovanella, Alfredo Campennì, Salvatore Cannavò, Angela Alibrandi

**Affiliations:** 1Department of Economics, University of Messina, 98122 Messina, Italy; azirilli@unime.it; 2Department of Human Pathology of Adults and Developmental Age “G. Barresi”, University of Messina, 98125 Messina, Italy; rmruggeri@unime.it (R.M.R.); cannavos@unime.it (S.C.); 3Department for Life Quality Studies, Alma Mater Studiorum, University of Bologna, 40126 Bologna, Italy; maria.barbalace2@unibo.it (M.C.B.); silvana.hrelia@unibo.it (S.H.); 4Clinic for Nuclear Medicine and Molecular Imaging, Imaging Institute of Southern Switzerland, Ente Ospedaliero Cantonale, 6500 Bellinzona, Switzerland; luca.giovanella@eoc.ch; 5Clinic for Nuclear Medicine, University Hospital of Zürich, 8091 Zürich, Switzerland; 6Unit of Nuclear Medicine, Department of Biomedical Sciences and Morphological and Functional Images, University of Messina, 98125 Messina, Italy; acampenni@unime.it

**Keywords:** oxidative stress, food regimes, non-parametric combination test

## Abstract

(1) Background: This paper aims to assess the existence of significant differences between two dietary regimes (omnivorous vs. semi-vegetarian) with reference to some oxidative stress markers (SOD, GPx, TRxR, GR, AGEs, and AOPPs) using non-parametric combination methodology based on a permutation test. (2) Methods: At the endocrinology unit of Messina University Hospital, two hundred subjects were asked to fill out a questionnaire about their dietary habits. None were under any pharmacological treatment. Using the NPC test, all comparisons were performed stratifying patients according to gender, age (≤40 or >40 years), BMI (normal weight vs. overweight), physical activity (sedentary vs. active lifestyle), TSH, FT4 levels in quartiles, and diagnosis of Hashimoto’s thyroiditis. We evaluated differences in oxidative stress parameters in relation to two examined dietary regimes (omnivorous vs. semi-vegetarian). (3) Results: The antioxidant parameters GPx and TRxR were significantly lower in subjects with an omnivorous diet than in semi-vegetarians, particularly in females, both age groups, subjects with normal weight, those not affected by Hashimoto’s thyroiditis, and both the sedentary and active lifestyle groups. Finally, the AGE and AOPP markers were significantly lower in semi-vegetarians. (4) Conclusion: Thanks to the NPC methodology, we can state that dietary patterns exert a significant influence on some oxidative stress parameters.

## 1. Introduction

Oxidative stress is a detrimental process that occurs when excess free radicals accumulate in tissues and cells due to the loss of balance between oxidant production and antioxidant defense mechanisms [1]. Free radicals (mainly reactive oxygen species (ROS)) are normal products of cell metabolism and are necessary for several cell processes when present at low levels. On the contrary, excess free radicals, due to their overproduction and/or inadequate removal, cause the oxidation of all macromolecules (proteins, lipids, and DNA), altering their structure and inhibiting their normal functions. These oxidative changes result in cell injury, apoptosis, and death and promote tissue inflammation, damage, and dysfunction [2,3].

This paper is structured as follows:-Section 1.1 presents the topic of oxidative stress in general and the possible links with dietary habits;-Section 1.2 refers to the scientific background, focusing on the main contributions related to oxidative stress parameters;-Section 2 describes the materials and methods; in detail, the sample is described, and the non-parametric combination (NPC) methodology is defined, with an overview of scientific papers that illustrate the different fields in which this methodology can be applied;-Section 3 illustrates the results of the application of NPC methodology; more specifically, we compared two dietary regimens (omnivorous vs. semi-vegetarian), stratifying patients according to some confounding variables, such as sex, age, BMI, TSH, FT4, diagnosis of Hashimoto’s thyroiditis, and physical activity;-Section 4 presents a discussion of the obtained results;-Section 5 describes the conclusions and possible future developments of the covered topic.

### 1.1. Oxidative Stress: Generality and Relationship with the Modern Lifestyle

Free radicals are highly unstable and reactive molecules with one or more unpaired electrons on the external atomic orbital, which rapidly react with other molecules to catch the missing electron, generating new free radicals in a chain reaction. They are normally produced in cells during numerous enzymatic and metabolic processes, for which they are often necessary [1]. Under physiological conditions, there is an equilibrium between the production and removal of free radicals called redox homeostasis. Multiple enzymatic and nonenzymatic defense systems and antioxidants operate to prevent ROS accumulation and counteract oxidative damage. Enzymatic antioxidants include the glutathione peroxidase (GPx)/glutathione reductase (GR) system and glutathione-S-transferases (GSTs), which represent the first-line defense against oxidants in almost every cell type with a tissue-specific distribution. The thioredoxin (TRx)/thioredoxin reductase (TRxR) system represents another important system for ROS detoxification. At the highest levels of oxidants, catalases (CAT) and superoxide dismutase (SOD) also contribute to the enzymatic degradation of free radicals. Nonenzymatic antioxidant defenses include endogenous molecules, such as mitochondrial uncoupling proteins, reduced glutathione (GSH), and transport proteins synthesized in the liver (ceruloplasmin, transferrin, albumin, etc.), as well as exogenous products, such as uric acid and vitamins E and C, which are mainly derived from one’s diet [1,2,3]. Among the exogenous compounds, selenium (Se) is also relevant for antioxidant defense since both GPx and TRx are selenoproteins, which contain a Se atom in their catalytic domain in the form of selenocysteine [4]. Inadequate Se supplementation may impair both the expression and enzymatic activity of the antioxidant enzymes GPx and TRx [3], whereas Se supplementation may help prevent the oxidative damage of thyroid cells, as demonstrated in in vitro models [5].

When free radicals are produced in excess, are not adequately removed, or both, a condition of oxidative stress (OS) occurs and causes cell damage, apoptosis, and death, and tissue inflammation [2,6]. Almost all cell types and tissues are susceptible to oxidative damage, although with tissue-specific differences [1,2,7].

A condition of OS may be the consequence of physiological processes (aging and cell senescence), lifestyle changes (reduced physical activity, alcohol abuse, smoking, etc.), exposure to environmental triggers (pollutants, radiations, drugs, chemicals from any sources, etc.), or dysmetabolic or dietary factors (overweight/obesity, impaired glucose tolerance, dysbiosis, reduced intake of antioxidants, excess food additives, etc.) [3,5,7].

The modern lifestyle, which is associated with unhealthy changes in dietary habits, reduced physical activity, psychological stress overload, and increased exposure to chemicals from different sources (pesticides, heavy metals, food additives, environmental pollutants, smoking, etc.), may favor the occurrence of a condition of OS, which in turn can contribute to the increasing burden of chronic diseases in industrialized societies [3,7].

In particular, changes in dietary habits have emerged as a driver of or a main contributor to many pathological conditions and are related to increased OS at the tissue and cellular levels.

In previous decades, diets rich in salt, refined sugars, and calories, fats, and proteins from animal sources, and low in fiber have become more common in the general population than plant-based dietary regimens rich in whole grains, fruits, and vegetables. Additionally, the consumption of processed and fast food has increased, along with the consequent exposure to food additives and preservatives, contaminants, and food-contacting (packaging) materials. This dietary regimen, the so-called Western-type diet (WTD), has harmful effects on health and may enhance the risk of chronic inflammatory disorders, both indirectly, through increasing fat mass and obesity, and directly, by altering the intestinal microbiota composition, dysregulating immune responses, and enhancing OS [3,7,8,9,10,11,12].

### 1.2. Scientific Background

OS has been involved in the pathogenesis of several inflammatory and immune-mediated disorders [1,2,3,7], such as cardiovascular diseases, diabetes, neurodegenerative diseases, cancer [7], and autoimmune diseases (AIDs), including thyroid AID [3,13,14,15]. In autoimmune thyroid disorders (AITD), excess ROS could induce the modification of tissue proteins, which become neo-antigens or may dysregulate the immune system, promoting the onset of an AITD in genetically predisposed individuals [16,17]. Moreover, excess ROS increases the proinflammatory state by enhancing the synthesis and release of cytokines, further contributing to tissue damage and disease progression [3,13].

The most common AITD at any age is represented by autoimmune thyroiditis, also referred to as Hashimoto’s thyroiditis (HT), which is the main cause of hypothyroidism in iodine-sufficient areas. HT prevalence in the general population is estimated at around 5%, and its incidence has increased significantly over the last few decades, mostly in industrialized countries, maybe as a consequence of changing environmental factors, as reported above [18].

A close and bidirectional relationship exists between thyroid autoimmunity and OS, as it has emerged from experimental and clinical studies [3]. On the one hand, excess ROS causes the oxidation of proteins (for instance, thyroglobulin), which become highly immunogenic and may promote autoimmune reactions in genetically predisposed individuals. On the other hand, once autoimmunity has been triggered, the infiltrating T and B lymphocytes develop a chronic inflammatory milieu in which ROS accumulate and exert a toxic effect on the surrounding cells, contributing to parenchymal destruction and disease progression [3,18]. Moreover, antioxidant defenses are often reduced/impaired in HT, even in euthyroidism, and do not counteract ROS overproduction [3,18]. Overall, human studies described increased OS in TH, even in euthyroidism [3]. Ates et al. reported a negative correlation between serum total antioxidant activity and anti-thyroperoxidase antibodies (TPO-Ab) [15], while Baser et al. reported a positive correlation between serum oxidants and anti-thyroglobulin (Tg-Ab) antibodies [19], and Ruggeri et al. confirmed that the TPO-Ab were independent predictors of the oxidative status in euthyroid HT patients [14].

As the role of OS in thyroid autoimmunity has emerged, interest has grown in peripheral/circulating biomarkers of oxidative stress, and many of these parameters have been measured to evaluate the impact and clinical relevance of oxidative stress in ATD, with sometimes conflicting results [3]. They include the following:-Antioxidant markers, such as antioxidant enzymes SOD, GR, GPx, TRxR, and paraoxonase 1 (PON-1) activity; total plasma antioxidant activity (TEAA); and biological antioxidant potential (BAP) [3,14,15,19,20,21,22].-Markers of oxidative stress, such as advanced glycation end products (AGEs) and their receptor (RAGE), advanced oxidation protein products (AOPPs), derived reactive oxygen (d-ROMS), malondialdehyde (MDA), oxidized-low density lipoprotein (ox-LDL), total oxidant status (TOS), and oxidative stress index (OSI), to mention a few [3,14,15,19,20,21,22,23,24,25,26].

In particular, our study group has provided strong evidence of the role of AGEs as a reliable marker of oxidative stress in HT [12,14,22,23]. AGEs are a family of compounds formed by nonenzymatic glycation of macromolecules (proteins, lipids, or nucleic acids). By binding their receptors (RAGEs) expressed on various cells, AGEs promote inflammation by regulating different signal transduction cascades and related transcription factors, such as nuclear factor kappa B (NF-kB). Two C-truncated isoforms of RAGE—the soluble RAGE and the esRAGE—bind to AGEs but fail to initiate an intracellular signal transduction cascade and, therefore, exert protective anti-inflammatory effects, acting as antagonists to the actions of AGEs [23].

The total body of AGEs’ burden is the sum of AGEs from dietary sources and endogenous synthesis. Methylglyoxal (MG) is the most common endogenous mediator of AGE synthesis, which is present ubiquitously in all cells as a product of normal metabolism, and its formation is markedly increased in hyperglycemia, accounting for the role of AGE accumulation in the pathogenesis of diabetic complications [27]. All other AGEs are exogenous, mainly derived from food. In particular, animal fat and proteins (mainly red meat and fatty cheese, followed by poultry, fish, and eggs) have higher levels of AGEs than carbohydrate foods, vegetables, and fruits [27]. Additionally, cooking techniques are important since roasting, broiling, and frying foods at high temperature increases their AGE content [27]. As a consequence, the WTD is high in AGEs. There is increasing evidence that AGEs play a role in the pathogenesis of several chronic diseases involving oxidative stress and inflammation, including cardiovascular, metabolic, and neurocognitive diseases, so reducing glycation load is certainly of health benefit [27].

We first reported higher serum levels of AGEs in HT patients than in healthy controls [14]. In the same studies, we assessed the oxidant/antioxidant status and found that oxidants (d-ROMs and AGEs) were increased and antioxidants (BAP) decreased in HT patients compared to controls [14]. Moreover, AGE serum levels were inversely correlated with serum antioxidant potential, indicating an imbalance between ROS production and antioxidant defenses in HT patients, i.e., a condition of OS [14]. We confirmed the finding of increased levels of AGEs in subsequent studies [12,22,23], correlating such an increase to reduced serum levels of the protective soluble receptor sRAGE [23] and reduced activity of the antioxidant paraoxonase (PON-1) [22]. Noteworthy, these changes in oxidative balance occurred irrespective of thyroid function alterations, autoimmunity per se being responsible for them.

The parameters mentioned above could be used as biochemical markers of oxidative stress and chronic inflammation to better predict disease onset and progression, allowing finer patient monitoring. Moreover, greater knowledge of the oxidative stress imbalance in chronic inflammatory disorders could open new perspectives in developing tailored therapeutic approaches to thyroid autoimmune disorders that significantly impair patients’ quality of life.

Moreover, in a recent study, we correlated changes in oxidative balance to the dietary habits of our HT patients. We reported that serum AGEs were significantly higher in HT patients than in controls, while the activity of GPx and TRxR, as well as total plasma antioxidant activity (TEAA), were lower, confirming a condition of oxidative stress in HT patients. When evaluating the dietary habits of our cohort of patients and controls, it emerged that nutritional patterns, mainly the consumption of animal foods, influenced the oxidative stress parameters in HT patients [12]. The results of this study suggested that a low intake of animal food (mainly meat) has a potentially protective effect on thyroid autoimmunity due to the positive influence of this dietary habit on redox balance.

Against this background, this paper’s main and original purpose is to evaluate, using a permutation-based statistical approach, the possible influence of dietary regimes on some oxidative stress markers.

## 2. Materials and Methods

### 2.1. NPC: Methodological Issue

We decided to apply the NPC methodology, based on permutation tests, because it allows us to perform comparison when the group sizes are unbalanced, and the asymptotic test may not be appropriate as the asymptotic distributions under the null hypothesis may be too far away from being appropriate; therefore, the NPC procedure represents a methodologically adequate solution to make an inference under these unbalanced conditions. A further reason that led us to use the NPC test is that this statistical methodology, free from distributional assumptions, allows us to perform stratified analyses.

As it is known, the Non-Parametric Combination (NPC) test represents a permutation solution for several multidimensional hypothesis-testing problems in the context of non-parametric inference [28,29,30,31].

NPC methodology has some advantages over the non-parametric ANOVA adjusted for multiple tests: firstly, it is of the multivariate type, allowing for the simultaneous testing of K variables of interest (MANOVA). In addition, assuming that the exchangeability of the data between the groups in the null hypothesis holds, it is characterized by two important properties:Property of similarity, as whatever the distribution underlying the data, the probability of rejection under the null hypothesis is invariant with respect to the set of data actually observed, and this is whatever the method of disclosure of the data;For any level of significance α, for any underlying distribution, and for all possible observed datasets, if under the alternative, the distribution dominates the null or is dominated by the null, then there is an undistorted conditional test, in the sense that the probability of rejection of the null hypothesis is always lower than the significance level α.

These two properties, respectively, guarantee that the conditional probability of rejecting the null hypothesis H_0_, when it is true, is always equal to the significance level α, regardless of how the data are detected, and that the conditional probability of rejecting H_0_, when the alternative H_1_ is true, is always not less than the chosen level of significance α—under the condition, however, that the data can be exchanged between groups. Thanks to these two characteristics, the inferences associated with permutation tests can be extended to the entire target population, respecting the properties of non-distortion and consistency [29,30].

The numerous properties that characterize this multivariate and multistratum methodology make it very flexible and widely applicable in various research contexts, including in the medical (endocrinological, hepatological, chronic intestinal diseases, etc.) and health (feeding, physical activity, disability, and epidemiology) fields [32,33,34]:-It is free from normality and homoscedasticity assumptions [35];-It can be effectively used even in the presence of small samples or in the presence of missing data [36,37];-It deals with variables of nominal, ordinal, and numerical nature without the need to identify the dependence structure among variables [38,39];-It allows for stratified analyses in which confounding factors can be isolated [40];

The NPC procedure verifies whether there are statistically significant differences between two or more groups of statistical units in relation to several variables. It involves two steps:-In the first step, the decomposition of a multivariate hypothesis system into one-dimensional sub-hypotheses occurs; for each sub-hypothesis, a partial permutation test is applied, enabling the examination of the marginal contribution of every variable in comparison between the groups (partial tests) [41,42];-In the second step, the non-parametric combination of partial tests allows us to obtain a second-order test called the multivariate hypothesis (combined tests) [43].

In the presence of a stratification variable, two levels of combination are defined: the first combines the partial tests into combined second-order tests, each corresponding to each stratum; the second pools the combined tests into a single third-order combined test.

Partial tests are non-parametrically merged through a Conditional Monte Carlo resampling procedure, using the Fisher, Tippett, or Liptak combination functions [29,30]; we cite current applications in the context of medical research for public health protection [44,45,46,47,48,49,50,51,52,53].

For all statistical analyses, the significance α level was set at 0.05.

The used statistical package was NPC test, version 2.0, Statistical Software for Multivariate Nonparametric Permutation Test, Copyright 2001, Methodologica s.r.l.

### 2.2. The Data

In this paper, we analyzed data obtained from a sample of subjects previously detected [12]. In particular, two hundred subjects were enrolled at the Endocrinology Unit of our University Hospital and were invited to compile a validated questionnaire about their dietary habits (Appendix A). Inclusion criteria at recruitment were to (i) be a Caucasian subject stably living in the city of Messina aged ≥ 18 years; (ii) have stable dietary habits in the last 5 years; and (iii) not be taking any pharmacological treatment and/or any form of supplements, to obtain a study population as homogeneous as possible. Exclusion criteria were BMI > 30 kg/m^2^; history of cancer or chronic cardiovascular, metabolic, or inflammatory disease; smoking; and alcohol abuse. All subjects provided written informed consent, expressing their authorization to the anonymous treatment of the information they had provided. The study was conducted according to the Declaration of Helsinki, and it was approved by the Ethics Committee of the University Hospital, AOU Policlinico “G. Martino” of Messina (study number 2017/19 Principal investigator: R.M. Ruggeri).

In a previous study [12], a comparison between subjects diagnosed with HT vs. healthy subjects was carried out. The present analysis was focused on the comparison between groups of subjects, defined according to their dietary habits. Thus, on the same sample, we compared omnivorous and semi-vegetarian subjects. In this sample, semi-vegetarians were defined as subjects on a plant-based diet, consisting mainly of vegetables and foods made from cereal grains, eggs, and dairy products, with occasional inclusion of fish (no more than twice a week) and meat (mainly white meat and poultry instead of red/processed meat, no more than once a week).

This paper aims to assess the existence/non-existence of significant differences between omnivorous (139 subjects) vs. semi-vegetarian subjects (61 subjects) with reference to some oxidative stress parameters. The following biomarkers have been measured on plasma samples from each study participant: the activity of the antioxidant enzymes SOD, GR, GPx, TRxR, and oxidant markers AGEs and AOPPs.

In our paper, we stratified for different confounding factors in order to isolate their effect: gender (male vs. female), age class (≤40 vs. >40 years), BMI class (normal weight vs. overweight), FT4 and TSH in the low vs. the high quartiles (quartile boundaries for TSH were <1.40, 1.41–1.98, 1.99–2.80, and ≥2.81 mU/L; quartile boundaries for FT4 were <14.46, 14.47–15.8, 15.9–17.1, and >17.32 pm/L), diagnosis of Hashimoto’s thyroiditis (yes or no), and physical activity (sedentary vs. active lifestyle). For THS and FT4, we merged the observations that fall within the first two quartiles (indicating them as low-normal) and those that fall within the last two quartiles (indicating them as high-normal).

Figure 1a shows the whole sample distribution according to these confounding factors using circle diagram; Figure 1b shows the association between food regimes (omnivorous and semi-vegetarian) and each confounding factor using bar chart; in particular, for each bar, we have reported the value of Chi-square test and the *p*-value, referred to the analysis of the statistical association between each confounding factor and the dietary regimes.

We specify that the imbalance in the male and female sample sizes derives from a different interest shown by the subjects when they joined the free screening (and therefore when completing the questionnaire on eating habits) during “International thyroid Awareness week”. Moreover, it is well known from the literature that thyroid disorders are more common in female subjects than in male subjects [18].

As we can see from the results shown in Figure 1b, there is a significant association between dietary regimes and some confounders, such as age class (66.2% of omnivores are under 40 years old), BMI class (88.5% of semi-vegetarians have a normal weight), FT4 class (62.6% of omnivores have a high-normal value of FT4), HT diagnosis (91.8% of semi-vegetarians do not have HT diagnosis), and physical activity (65.6% of semi-vegetarians have an active lifestyle).

## 3. Results

The application of the NPC test was performed considering the OS parameters as interest variables, the dietary regimes of omnivores (Omniv.) and Semi-vegetarians (Semi-Veg.) as grouping variables, and confounding factors (gender, age classes, etc.) as stratification variables.

In the hypotheses system, the distributional equality stated by H_0_ implies that the observed data vectors are exchangeable between two groups. Without loss of generality, we suppose that for each sub-hypothesis H_0i_ against H_1i_, there is a suitable partial permutation test assumed to be significant. The system of hypotheses is set in such a way that the related partial tests are jointly processed so that they can be non-parametrically combined by considering their underlying dependence structure within the non-parametric combination method. Our hypotheses system, which verifies whether there are overall differences between the multivariate distributions of the group outcomes, is the following:$$H_{0i}:\;\{{\mathrm{SOD}}_{1i}\overset{d}{=}{\mathrm{SOD}}_{2i}\}\cap^{\;}\ldots \cap^{\;}\left\{{\mathrm{AOPPs}}_{1i}\overset{d}{=}{\mathrm{AOPPs}}_{2i}\right\}$$
$$H_{1i}:\;\{{\mathrm{SOD}}_{1i}\overset{d}{\neq }{\mathrm{SOD}}_{2i}\}\cup^{\;}\ldots \cup^{\;}\left\{{\mathrm{AOPPs}}_{1i}\overset{d}{\neq }{\mathrm{AOPPs}}_{2i}\right\}$$
where $\overset{d}{=}$ refers to the equal distribution symbol; 1 and 2 are the dietary regimes, omniv. vs. semi-veg.; and i is the stratification index, referring to seven confounding factors, which assume the values 1 and 2 in each stratification: gender (male vs. female), age class (≤40 vs. >40 years), BMI class (≤25 vs. >25), FT4 (low-normal vs. high-normal), TSH (low-normal vs. high-normal), Hashimoto’s thyroiditis (yes or no), and physical activity (sedentary or active lifestyle). The global H_0i_ hypothesis is true if all sub-nulls are jointly true (for intersection criterion), and the alternative hypothesis H_0i_ is true when at least one sub-alternative is true (for union criterion).

In Table 1, Table 2, Table 3, Table 4, Table 5, Table 6 and Table 7, we report mean ± standard deviations (SD), median, and, in brackets, the first quartile Q1 and the third quartile Q3 of each oxidative stress parameter for omnivorous and semi-vegetarian subjects, in order to better describe the distribution of the data, which is exactly what permutation methods define and use; the last column reports the partial and combined *p*-values obtained by the application of the NPC test. In bold, we report the significant *p*-values (<0.05). The ↓ symbol indicates the combination of the *p*-values of the partial tests into a single *p*-value of the combined test.

Examining the results of the comparison between omnivores and semi-vegetarians, obtained using NPC methodology, we can see that there are statistically significant differences, such as resulting from significant combined *p*-values in several strata: more specifically, in subjects with ages higher than 40 years (*p* = 0.009, Table 2), with normal weight (*p* = 0.005, Table 3), with low-normal FT4 levels (*p* = 0.037, Table 5), and without HT diagnosis (*p* = 0.026, Table 6).

Focusing attention on the partial test, we can observe many partially significant *p*-values, especially regarding two OS parameters: the antioxidant enzymes GPx and TRxR. In particular, they are significantly higher in semi-vegetarians compared with omnivores in female subjects (Table 1); in subjects of both age groups (≤40 years and >40 years) (Table 2); in subjects with normal weight (BMI ≤ 25) (Table 3); in subjects with both low-normal and high-normal TSH levels (Table 4) and low-normal FT4 levels (Table 5); in subjects not affected by Hashimoto’s thyroiditis (Table 6); and in subjects who are either physically active or sedentary (Table 7). We also found that AGEs were significantly lower in semi-vegetarians over the age of 40 (Table 2). In addition, within the strata of subjects with normal weight (Table 3) and low-normal FT4 levels (Table 5), semi-vegetarians show significantly lower AOPPs values than omnivores.

Figure 2, Figure 3, Figure 4 and Figure 5, show dotplots, with reference to GPx and TRxR parameters, referring to some significant comparisons between omnivorous and semi-vegetarian subjects. Among all the statistically significant results deriving from the NPC test application, we realized the dotplots only for the pair “GPx and TRxR” because they are the parameters for which significance occurs most frequently. More specifically, we plotted this pair of variables only if their significance occurred jointly and only for a single specific stratum in each stratification, in order to show only results that are attributable to a specific stratum; this allows us to identify confounding factors monitored through stratification.

## 4. Discussion

In recent years, interest in the impact of nutrition on health outcomes has grown as the incidence/prevalence of overweight/obesity and chronic non-communicable diseases has risen in the general population, mostly in developed countries. Unhealthy dietary habits, high in calories and low in nutritional value, contribute to such an increase, and a deeper understanding of the underlying mechanisms could help prevent these disorders and avoid complications.

The role of nutrition in the development and prevention of chronic disorders such as metabolic syndrome, diabetes, and cardiovascular disease has long been known. In the past decade, this impact has also emerged in various autoimmune conditions [7,8,9]. In particular, as the incidence of HT and related dysfunction has increased in industrialized countries, research has shed new light on the complex interplay between diet and thyroid autoimmunity. On the one hand, the nutritional needs of patients suffering from HT have been discussed in the literature, with still inconclusive and sometimes conflicting results [54,55,56,57,58,59,60,61,62]. On the other hand, a protective or predisposing role of specific dietary traits has been investigated to propose a healthy diet model [12,61,62,63,64,65]. Indeed, current options for preventive interventions in subjects at risk of developing AITD are still very limited.

In this setting, special attention has been paid to the so-called “Western lifestyle”, characterized by excessive caloric intake; a diet rich in fats, refined sugars, and salt, and low in fibers, fruit, and vegetables (the WTD); increased consumption of processed and fast foods; and reduced physical activity. Adherence to such a lifestyle by an increasing number of people, mostly low-income and young age groups, has contributed to the rise in the incidence of several diseases of “modern society” linked to a chronic inflammatory state [66]. This unhealthy dietary pattern may promote autoimmunity onset/progression through several mechanisms: disruption of gut microbiota and altered intestinal permeability (the so-called leaky gut syndrome); hormonal effects (resistin and adiponectin/leptin ratio); modulation of immune response (altered Treg/TH17 balance) and upregulation of inflammatory cytokines (IL-6 and TNFἀ); low-grade systemic inflammation (the so-called metainflammation); and lack of exogenous antioxidants (from fruit and vegetables) [66,67,68].

As opposed to the WTD, the Mediterranean diet (MD) is characterized by a high intake of vegetables, legumes, fresh fruits, nuts, whole grains, and olive oil; moderate intake of fish (mainly small oily fishes), dairy products, and poultry; and low consumption of eggs, red meat, and processed meat products [69]. This dietary regimen, low in salt and refined sugars and high in natural antioxidants, vitamins, and fibers, favors gut microbiota symbiosis and exerts antioxidants and anti-inflammatory effects. For this reason, the MD, named after the traditional dietary regimens of populations living in the Mediterranean basin, has been proposed as a healthy food model [69].

Several studies have investigated the intriguing link between nutrition and health issues in the setting of AIDs, including rheumatoid arthritis, multiple sclerosis, psoriasis, intestinal bowel diseases, and HT, consistently indicating MD traits as protective against autoimmunity, as opposed to the WTD [65,70,71,72,73,74].

We focused our research on the effect of these dietary patterns on oxidative stress as a key feature of thyroid autoimmunity.

In a previous study, we investigated the nutritional habits of a cohort of euthyroid HT patients compared to healthy controls in relation to the oxidant/antioxidant balance [12]. Our HT patients reported a higher intake of animal products, saturated fats, and refined sugars and a lower intake of fiber, fruits, and vegetables than healthy subjects, who, in turn, reported a higher intake of plant foods. Thus, HT’s nutritional pattern resembled the WTD, while controls displayed a higher level of adherence to the MD [12]. In this light, our study provided data in favor of the protective role of MD against thyroid autoimmunity, whereas excess consumption of meat and animal products appeared to represent a predisposing factor [12]. Another important finding was the influence of nutritional patterns on oxidative stress parameters. Indeed, in our study population, the intake of animal foods, mostly meat, significantly affected the levels of both oxidants (AGEs) and antioxidants (GPx, TRxT, and TEAA), increasing the former and lowering the latter [12].

To further strengthen these findings, we evaluated our data using a different statistical approach, subdividing the subjects under study into two groups according to their dietary habits and stratifying them by different factors. We found that antioxidants (GPx and TRxR) were increased and oxidants (AGEs and AOPPs) decreased in semi-vegetarians compared with omnivores, providing further evidence of the effect of different dietary patterns on redox balance. A plant-based dietary regimen was associated with increased antioxidant defenses in subjects of both age groups, mainly those with a normal BMI. Additionally, oxidants were lower in semi-vegetarians than in omnivores, mainly in subjects over 40, suggesting that a plant-based diet may help counteract the increase in OS with increasing age. Furthermore, dietary traits were more relevant than physical activity in influencing the oxidative balance, as well as the appearance of thyroid autoimmunity, since no significant differences between the groups (omnivorous vs. semi-vegetarian) emerged when stratifying them according to the levels of physical activity (active lifestyle or sedentary).

Finally, the present statistical analysis confirmed the previous finding of increased oxidative stress in subjects suffering from HT, irrespective of TSH values, and further pointed to animal products of omnivorous diets as the main nutritional factors associated with increased risk of thyroid autoimmunity.

In summary, our research focused on the relationship between nutrition and thyroid autoimmunity shed new light on important processes in HT pathogenesis (namely, oxidative stress), and, though the recognition of specific food-disease interactions, may contribute to the development of tailored dietary regimens for HT patients, as well as dietary guidelines for populations at risk, enabling a possibility of prevention of autoimmunity. On this basis, the recommended diet for HT patients may be the MD, an anti-inflammatory and antioxidant dietary pattern that also provides the appropriate supply of iodine, selenium, iron, and vitamin D, which are found in plant products rich in polyphenols, antioxidants, and omega-3 fatty acids [69].

Limitations of the present study were the relatively small number of recruited subjects and the rather high prevalence of female subjects in the study group. Indeed, it is well known that sex differences exist in OS stress mechanisms since males display higher production of ROS and less-efficient antioxidant mechanisms, so the existence of gender differences in oxidative and inflammatory markers should be taken into account [75]. Consequently, this study group cannot be considered representative of the general population, and further studies on a novel and large population are needed to analyze the impact of diet on oxidative stress and related disorders more in depth.

Finally, we acknowledge the limitations of double data analysis since we have re-evaluated data that have already been analyzed using a different statistical approach. However, we tried to avoid circular analysis bias by modifying the classification parameters (the presence of a TH in the previous analysis and the dietary habits in the present one).

## 5. Conclusions

The present research paper, based on the application of the multivariate and multistratum NPC methodology, allowed us to identify some interesting relationships between oxidative stress parameters and dietary habits. More specifically, oxidative stress parameters were compared between omnivores and semi-vegetarians, stratified for some confounders considered relevant for the analysis.

The obtained results show that, globally, considering all the parameters of oxidative stress, omnivores and semi-vegetarians show statistically significant differences in several subgroups: in subjects older than 40, subjects with normal weight, subjects with low-normal FT4 levels, and subjects without HT diagnosis.

More specifically, among all examined OS parameters, the antioxidants GPx and TRxR play a particular role in discriminating omnivores and semi-vegetarians and were significantly higher in semi-vegetarians, specifically in females, subjects with normal weight, subjects with low-normal FT4 values, and subjects not affected by HT. GPx and TRxR are significantly higher in semi-vegetarians and in all strata defined by age, TSH, and physical activity.

As a further result, we found that AGEs are significantly lower in semi-vegetarians over 40 years old. It is known that high values of AGEs have negative effects, as they favor the development of certain pathologies affecting the metabolic system, for which the semi-vegetarian diet, in subjects over 40, would seem to have a protective effect.

Therefore, in light of the results, we can say that changing eating habits, mostly preferring foods of plant origin, can be a useful strategy to counteract oxidative stress. Unhealthy eating habits, including inadequate intake of vegetables and fruits, may lead to nutritional deficiencies, intestinal dysbiosis, chronic systemic inflammation, and reduced antioxidant potential, and represents risk factors for several chronic disorders of “modern society”, including HT. Improving these aspects of nutrition ameliorates the healthy status of HT patients, supports medical therapies, and may also protect subjects at risk of developing the disease.

The future developments of this research include a comparison between three groups of subjects (omnivores, semi-vegetarians, and vegans) in order to investigate more deeply the influence of dietary habits on oxidative stress parameters on a higher number of enrolled subjects.

## Figures and Tables

**Figure 1 healthcare-11-02263-f001:**
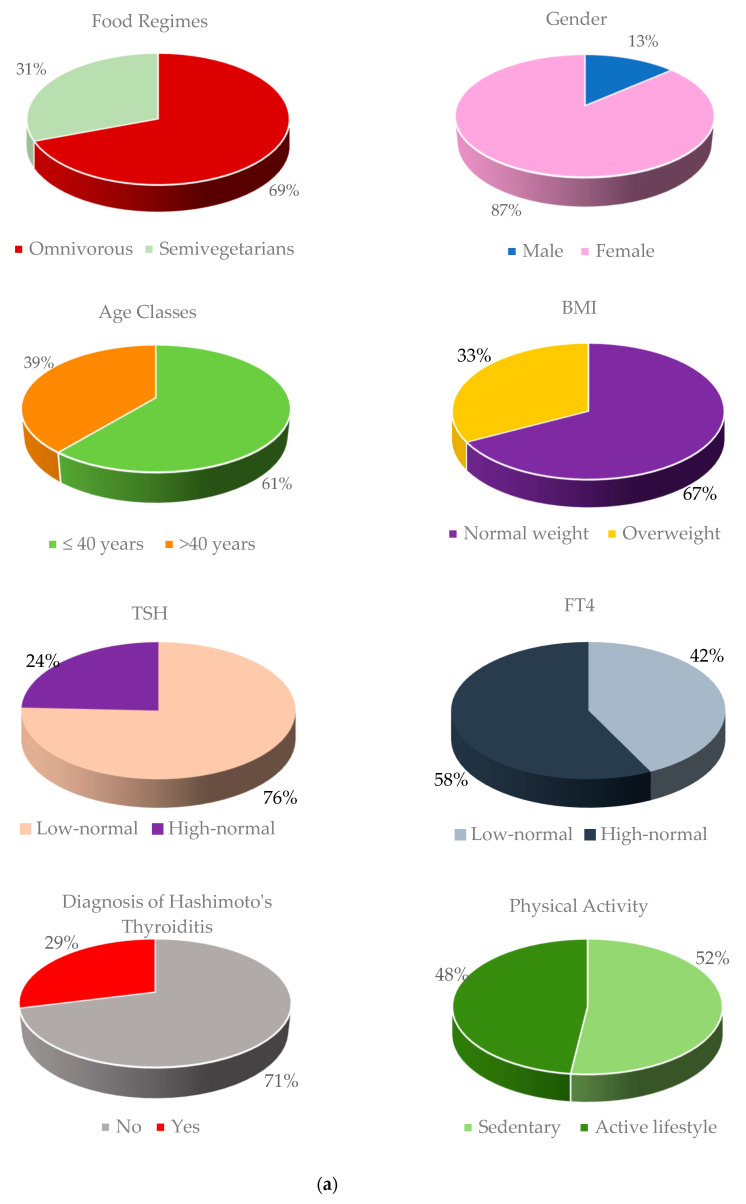
(**a**) Distribution of the whole sample for food regimes and confounding factors; (**b**) Association between food regimes (omnivorous and semi-vegetarian) and confounding factors.

**Figure 2 healthcare-11-02263-f002:**
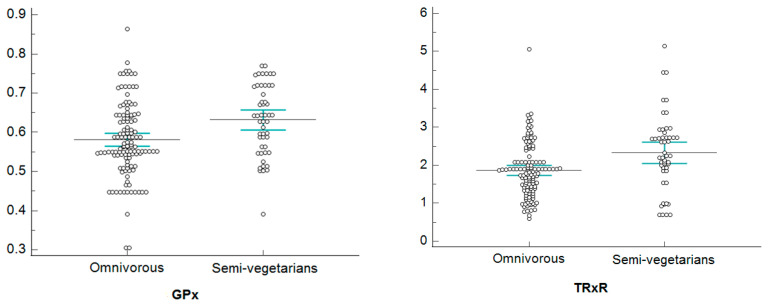
Dotplots of GPx and TRxR—female stratum.

**Figure 3 healthcare-11-02263-f003:**
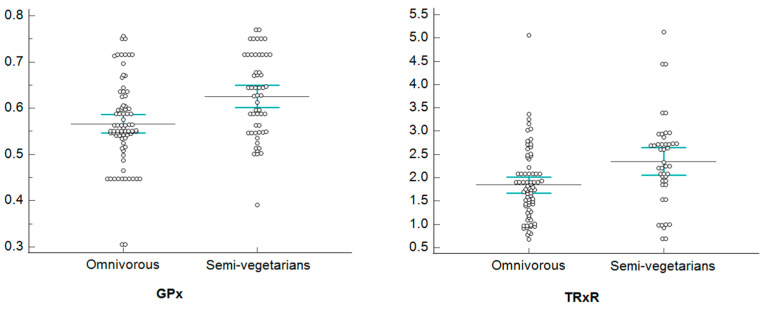
Dotplots of GPx and TRxR—normal weight stratum.

**Figure 4 healthcare-11-02263-f004:**
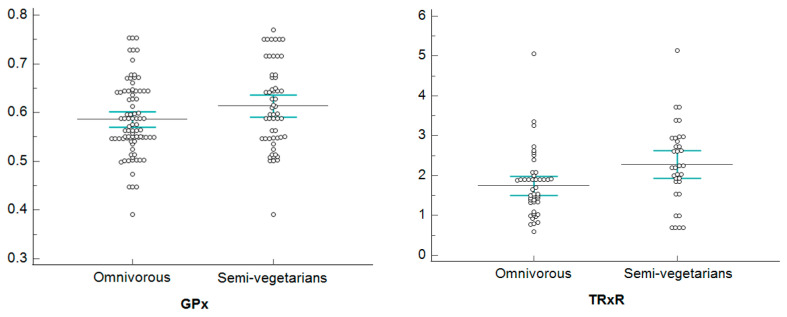
Dotplots of GPx and TRxR—low-normal FT4 stratum.

**Figure 5 healthcare-11-02263-f005:**
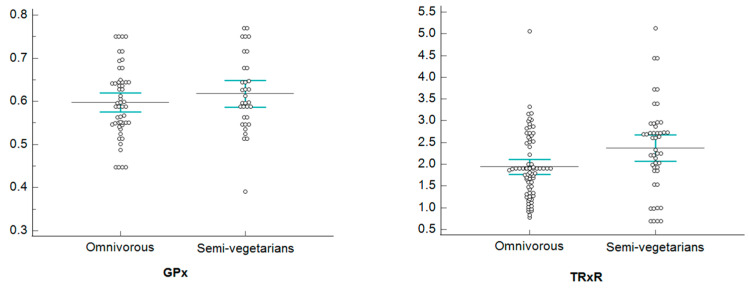
Dotplots of GPx and TRxR—no Hashimoto’s thyroiditis stratum.

**Table 1 healthcare-11-02263-t001:** Mean ± SD, median and quartiles (Q1–Q3), and *p*-value of comparison between omnivores and semi-vegetarians for oxidative stress parameters stratified by gender.

Stratum 1: Male				Stratum 2: Female		
Variables	Omniv.	Semi-Veg.	*p*-Value	Omniv.	Semi-Veg.	*p*-Value
SOD	4.92 ± 0.86	4.74 ± 0.42	0.535	4.87 ± 0.96	4.86 ± 0.74	0.960
	5.32 (4.22–5.67)	4.59 (4.32–4.90)		4.77 (4.17–5.68)	4.90(4.42–4.51)	
GPx	0.58 ± 0.01	0.62 ± 0.01	0.068	0.58 ± 0.10	0.63 ± 0.01	**0.016**
	0.58 (0.54–0.64)	0.64 (0.58–0.70)		0.59 (0.52–0.65)	0.61 (0.56–0.66)	
TRxR	2.17 ± 0.87	2.55 ± 0.48	0.202	1.87 ± 0.71	2.33 ± 0.99	**0.001**
	1.97 (1.45–2.41)	2.72 (2.08–2.91)		1.86 (1.34–2.22)	2.25 (1.85–2.76)	
GR	70.30 ± 21.95	63.84 ± 19.53	0.459	70.29 ± 16.68	69.62 ± 12.15	0.800
	72 (57.6–84.7)	64.39 (48.06–74.93)		71.52 (65.12–74.93)	70.39 (63.51–78.61)	
AGEs	152.5 ± 50.04	126.3 ± 30.82	0.104	126.5 ± 51.01	122.4 ± 41.66	0.606
	154.3 (111.6–192.2)	127.4 (114.8–151.1)		125.2 (107.2–135.8)	123.4 (91.6–134.4)	
AOPPs	1.18 ± 0.32	1.10 ± 0.01	0.455	1.06 ± 0.25	1.08 ± 0.28	0.640
	1.17 (1.02–1.36)	1.08 (1.06–1.09)		1.03 (0.93–1.07)	1.07 (0.95–1.13)	
**Combined** ***p*-value**			↓			↓
			0.380			0.132

**Table 2 healthcare-11-02263-t002:** Mean ± SD, median and quartiles (Q1–Q3), and *p*-value of comparison between omnivores and semi-vegetarians for oxidative stress parameters stratified by age class.

Stratum 1: Age ≤ 40 years				Stratum 2: Age > 40 Years		
Variables	Omniv.	Semi-Veg.	*p*-Value	Omniv.	Semi-Veg.	*p*-Value
SOD	4.80 ± 1.01	4.77 ± 0.69	0.853	5.01 ± 0.83	4.91 ± 0.70	0.576
	4.74 (4.16–5.49)	4.78 (4.38–5.31)		5.24 (4.43–5.78)	4.90 (4.43–5.51)	
GPx	0.57 ± 0.11	0.62 ± 0.01	**0.026**	0.60 ± 0.01	0.65 ± 0.01	**0.030**
	0.57 (0.53–0.64)	0.61 (0.56–0.64)		0.59 (0.52–0.64)	0.64 (0.59–0.74)	
TRxR	1.91 ± 0.78	2.40 ± 0.93	**0.005**	1.88 ± 0.65	2.34 ± 0.93	**0.012**
	1.88 (1.30–2.19)	2.59 (1.95–2.91)		1.90 (1.39–2.44)	2.21 (1.71–2.76)	
GR	72.31 ± 15.06	71.53 ± 10.69	0.789	66.32 ± 20.44	65.76 ± 15.84	0.890
	71.52 (67.26–75.72)	70.63 (66.52–77.61)		71.52 (53.99–74.93)	70.39 (50.49–81.58)	
AGEs	118.8 ± 43.02	119.9 ± 38.89	0.899	149.9 ± 59.60	126.2 ± 36.89	**0.042**
	114.8 (99.75–127.4)	120.7 (81.9–127.4)		145.8 (118.1–186.1)	127.4 (114.8–186.6)	
AOPPs	1.12 ± 0.34	1.03 ± 0.23	0.096	1.15 ± 0.30	1.05 ± 0.14	0.063
	1.11 (0.95–1.17)	1.03 (0.91–1.07)		1.14 (1.03–1.43)	1.07 (1.03–1.26)	
**Combined** ***p*-value**			↓			↓
			0.250			**0.009**

**Table 3 healthcare-11-02263-t003:** Mean ± SD, median and quartiles (Q1–Q3), and *p*-value of comparison between omnivores and semi-vegetarians for oxidative stress parameters stratified by BMI class.

Stratum 1: Normal Weight				Stratum 2: Overweight		
Variables	Omniv.	Semi-Veg.	*p*-Value	Omniv.	Semi-Veg.	*p*-Value
SOD	5.04 ± 0.96	4.92 ± 0.65	0.456	4.65 ± 0.91	4.22 ± 0.70	0.227
	5.01 (4.37–5.75)	4.90 (4.43–5.47)		4.62 (3.84–5.42)	4.33 (3.23–4.85)	
GPx	0.57 ± 0.01	0.62 ± 0.01	**0.006**	0.63 ± 0.01	0.67 ± 0.01	0.302
	0.59 (0.52–0.64)	0.61 (0.56–0.65)		0.62 (0.58–0.69)	0.65 (0.64–0.75)	
TRxR	1.88 ± 0.81	2.40 ± 0.89	**0.001**	1.90 ± 0.62	2.15 ± 1.24	0.381
	1.79 (1.29–2.31)	2.62 (1.90–2.88)		1.91 (1.38–2.36)	2.08 (0.71–3.72)	
GR	72.90 ± 16.65	67.94 ± 13.31	0.071	66.64 ± 17.52	73.50 ± 17.19	0.334
	71.52 (66.53–76.73)	69.39 (62.41–77.27)		71.52 (58.68–74.93)	75.43 (50.49–91.87)	
AGEs	123.9 ± 35.08	126.1 ± 38.17	0.735	136.9 ± 67.53	100.2 ± 25.53	0.140
	124.1 (119.4–127.4)	125.2 (109.6–135.3)		137.4(105.3–162.3)	91.12 (78.28–127–4)	
AOPPs	1.06 ± 0.20	0.99 ± 0.16	**0.047**	1.27 ± 0.52	1.17 ± 0.33	0.495
	1.07 (1.06–1.09)	1.03 (0.88–1.07)		1.28 (0.82–2.02)	1.16 (1.02–1.34)	
**Combined** ***p*-value**			↓			↓
			**0.005**			0.336

**Table 4 healthcare-11-02263-t004:** Mean ± SD, median and quartiles (Q1–Q3), and *p*-value of comparison between omnivores and semi-vegetarians for oxidative stress parameters stratified by TSH class.

Stratum 1: Low-Normal				Stratum 2: High-Normal		
Variables	Omniv.	Semi-Veg.	*p*-Value	Omniv.	Semi-Veg.	*p*-Value
SOD	4.84 ± 1.03	4.49 ± 0.75	0.829	5.00 ± 0.68	4.78 ± 0.39	0.352
	4.77 (4.28–5.65)	4.50 (4.34–5.46)		4.98 (4.08–5.79)	4.85 (4.61–5.32)	
GPx	0.59 ± 0.01	0.63 ± 0.01	**0.044**	0.59 ± 0.01	0.62 ± 0.01	**0.047**
	0.58 (0.54–0.64)	0.61 (0.56–0.68)		0.55 (0.51–0.66)	0.61 (0.56–0.67)	
TRxR	1.89 ± 0.77	2.37 ± 0.97	**0.005**	1.92 ± 0.61	2.47 ± 0.77	**0.028**
	1.89 (1.37–2.11)	2.23 (1.85–2.72)		1.84 (1.36–2.51)	2.93 (2.41–3.18)	
GR	71.46 ± 16.99	71.40 ± 44.20	0.984	66.66 ± 17.63	60.32 ± 14.85	0.334
	71.52 (64.09–74.93)	70.39 (63.88–78.28)		67.52 (64.79–74.93)	61.0 (53.7–69.9)	
AGEs	125.8 ± 51.04	128.7 ± 37.62	0.776	140.1 ± 51.64	105.5 ± 34.57	0.066
	126.4 (108.2–148.2)	127.8 (116.1–128.4)		127.4 (78.28–138.18)	114.8 (111.2–127.4)	
AOPPs	1.12 ± 0.25	1.03 ± 0.22	0.076	1.19 ± 0.34	0.96 ± 0.18	0.052
	1.11 (1.04–1.15)	1.03 (0.92–1.07)		1.17 (0.82–1.29)	1.03 (0.98–1.07)	
**Combined** ***p*-value**			↓			↓
			0.053			0.059

**Table 5 healthcare-11-02263-t005:** Mean ± SD, median and quartiles (Q1–Q3), and *p*-value of comparison between omnivores and semi-vegetarians for oxidative stress parameters stratified by FT4 class.

Stratum 1: Low-Normal				Stratum 2: High-Normal		
Variables	Omniv.	Semi-Veg.	*p*-Value	Omniv.	Semi-Veg.	*p*-Value
SOD	4.69 ± 0.96	4.99 ± 0.66	0.120	4.98 ± 0.94	4.66 ± 0.69	0.094
	4.74 (3.85–5.41)	5.07 (4.37–5.51)		4.93 (4.28–5.78)	4.71 (4.43–5.23)	
GPx	0.58 ± 0.01	0.63 ± 0.01	**0.012**	0.59 ± 0.11	0.63 ± 0.11	0.125
	0.59 (0.53–0.63)	0.61 (0.59–0.65)		0.60 (0.56–0.70)	0.64 (0.52–0.66)	
TRxR	1.95 ± 0.74	2.39 ± 0.93	**0.019**	1.93 ± 0.79	2.32 ± 0.94	0.056
	1.91 (1.40–2.32)	2.25 (1.85–2.72)		1.81 (1.34–2.34)	2.69 (2.08–2.94)	
GR	68.89 ± 15.93	68.97 ± 12.67	0.982	71.12 ± 18.00	78.12 ± 15.15	0.422
	71.52 (64.86–72.53)	70.39 (62.41–78.61)		71.52 (63.49–77.06)	76.07 (56.87–77.27)	
AGEs	120.4 ± 39.13	124.4 ± 27.86	0.609	134.7 ± 57.00	121.5 ± 47.27	0.266
	119.5 (101.1–127.4)	124.7 (120.3–128.2)		127.4 (114.9–149.8)	122.4 (77.99–127–4)	
AOPPs	1.11 ± 0.18	1.01 ± 0.19	**0.031**	1.10 ± 0.29	1.06 ± 0.33	0.451
	1.10 (0.96–1.19)	1.03 (0.88–1.07)		1.08 (0.96–1.13)	1.07 (0.82–1.27)	
**Combined** ***p*-value**			↓			↓
			**0.037**			0.130

**Table 6 healthcare-11-02263-t006:** Mean ± SD, median and quartiles (Q1–Q3), and *p*-value of comparison between omnivores and semi-vegetarians for oxidative stress parameters stratified by diagnosis of Hashimoto’s thyroiditis.

Stratum 1: No HT Diagnosis				Stratum 2: Yes HT Diagnosis		
Variables	Omniv.	Semi-Veg.	*p*-Value	Omniv.	Semi-Veg.	*p*-Value
SOD	4.82 ± 1.08	4.87 ± 0.71	0.782	4.97 ± 0.71	4.56 ± 0.28	0.217
	4.78 (3.87–5.68)	4.90 (3.34–5.46)		4.88 (4.42–5.60)	(4.43 (4.43–4.75)	
GPx	0.58 ± 0.01	0.61 ± 0.01	**0.036**	0.64 ± 0.01	0.65 ± 0.11	0.851
	0.59 (0.52–0.65)	0.61 (0.57–0.64)		0.64 (0.56–0.74)	0.64 (0.53–0.74)	
TRxR	1.96 ± 0.77	2.42 ± 0.94	**0.003**	1.80 ± 0.67	1.85 ± 0.52	0.880
	1.91 (1.34–2.89)	2.66 (1.87–2.91)		1.82 (1.39–2.08)	1.90 (1.51–2.08)	
GR	71.51 ± 16.95	68.19 ± 14.25	0.219	68.30 ± 17.66	73.00 ± 4.32	0.562
	71.53 (65.01–74.02)	69.39 (59.60–78.61)		74.93 (58.29–74.93)	73.93 (70.11–74.93)	
AGEs	114.7 ± 42.4	124.3 ± 37.64	0.168	153.0 ± 56.02	109.5 ± 39.97	0.094
	114.9 (94.23–127.4)	124.2 (114.7–127.4)		147.3 (127.4–187.6)	110.7(82.73–117.8)	
AOPPs	1.09 ± 0.25	1.04 ± 0.27	0.269	1.12 ± 0.28	1.01 ± 0.15	0.321
	1.07 (0.94–1.15)	1.03 (0.92–1.07)		1.07 (1.01–1.20)	1.02 (0.90–1.08)	
**Combined** ***p*-value**			↓			↓
			**0.026**			0.614

**Table 7 healthcare-11-02263-t007:** Mean ± SD, median and quartiles (Q1–Q3), and *p*-value of comparison between omnivores and semi-vegetarians for oxidative stress parameters stratified by physical activity.

Stratum 1: Sedentary				Stratum 2: Active Lifestyle		
Variables	Omniv.	Semi-Veg.	*p*-Value	Omniv.	Semi-Veg.	*p*-Value
SOD	4.87 ± 0.95	4.71 ± 0.48	0.451	4.88 ± 0.96	4.91 ± 0.78	0.896
	4.88 (4.16–5.74)	4.67 (4.43–4.84)		4.78 (4.22–5.49)	4.95 (4.31–5.51)	
GPx	0.58 ± 0.01	0.63 ± 0.01	**0.041**	0.58 ± 0.10	0.63 ± 0.11	**0.043**
	0.59 (0.54–0.63)	0.61 (0.56–0.66)		0.59 (0.52–0.66)	0.61 (0.56–0.68)	
TRxR	1.96 ± 0.66	2.39 ± 0.93	**0.018**	1.81 ± 0.83	2.36 ± 0.94	**0.003**
	1.90 (1.52–2.46)	2.33 (1.89–2.72)		1.71 (1.31–2.02)	2.60 (1.87–2.91)	
GR	68.92 ± 18.17	68.61 ± 14.30	0.941	72.31 ± 15.69	68.56 ± 13.64	0.225
	71.53 (62.02–74.93)	70.39 (64.59–76.77)		71.53 (66.83–75.72)	70.39 (58.66–80.50)	
AGEs	126.0 ± 47.18	116.9 ± 25.34	0.398	134.2 ± 57.11	126.3 ± 42.74	0.456
	127.3 (101.5–149.5)	126.6 (101.1–127.6)		129.1 (81.78–131.8)	127.1 (114.8–135.4)	
AOPPs	1.06 ± 0.24	1.01 ± 0.14	0.426	1.09 ± 0.30	1.12 ± 0.30	0.635
	1.03 (0.92–1.07)	1.02 (0.91–1.22)		1.03 (0.98–1.07)	1.11 (0.96–1.18)	
**Combined** ***p*-value**			↓			↓
			0.479			0.112

## Data Availability

Data available on request from the authors.

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
