# Peer review of "The Influence of Food Regimes on Oxidative Stress: A Permutation-Based Approach Using the NPC Test"

_healthcare, 2023, doi:10.3390/healthcare11162263_

Round 1

Reviewer 1 Report

The subject discussed is a peculiar public health issue and contents of the research are very well presented. Overall, the research manuscript is of good quality.
 My specific comments are as follows
Lines 62-78 are redundant as such fashion is adopted in report/thesis writing and this manuscript is still complete without these lines.
Lines 228-30, please specify the sub-areas rather than mentioning health and medical field in general. 
Line 259, "casuistry" this term needs elaboration if it is an appropriate one to be used in this manuscript as it has been used more than once in this document.
Line 261 tells that enrolled subjects were invited to compile a validated questionnaire, so by whom were the compilation done, what process was adopted for compilation as well as validation of the tool. This methodology is devoid of matter on the tool used. 
Line 280, existence should be replaced by existence/non-existence.
Lines 285-91 should be shifted under the 2.1 Heading as it is more suitable there.
1st three paragraphs of the conclusion (lines 486-502) are repetitive and should be concise or rephrased as these are not a conclusion.  

Author Response

The subject discussed is a peculiar public health issue and contents of the research are very well presented. Overall, the research manuscript is of good quality.
 My specific comments are as follows:

  • Lines 62-78 are redundant as such fashion is adopted in report/thesis writing and this manuscript is still complete without these lines.

Lines 62-78 present the structure of the paper and we authors generally  prefer to draw up this outline to facilitate reading and better understanding by readers. We are aware that these lines seem redundant, but we sincerely believe they are useful.

  • Lines 228-30, please specify the sub-areas rather than mentioning health and medical field in general. 

We thank the referee and in lines 228-230 we specified the sub-areas that refer to the health and medical fields. In the new versions of our manuscript we specified that: “…various research contexts, including in the medical (endocrinological, hepatological, chronic intestinal diseases, etc) and health (feeding, physical activity, disability, epidemiology) fields”.

  • Line 259, "casuistry" this term needs elaboration if it is an appropriate one to be used in this manuscript as it has been used more than once in this document.

In lines 259 (now 260) we have replaced the term “casuistry” with the expression “sample of subjects previously detected”.

  • Line 261 tells that enrolled subjects were invited to compile a validated questionnaire, so by whom were the compilation done, what process was adopted for compilation as well as validation of the tool. This methodology is devoid of matter on the tool used. 

Line 261: we thank the Referee for this very pertinent comment, that gives us the opportunity to explain how data were collected. We have omitted this relevant methodological information in the Methods section because specified in the previous study [ref 12]. We have now provided this additional information as “supplementary material” (new attached file: “Supplementary material”).

  • Line 280, existence should be replaced by existence/non-existence.

In line 280 (now281) we added “existence/no existence”, as you suggested.

  • Lines 285-91 should be shifted under the 2.1 Heading as it is more suitable there.

The lines 285-291 have been moved to the beginning of 2.1 section,  as you suggested.

  • 1t three paragraphs of the conclusion (lines 486-502) are repetitive and should be concise or rephrased as these are not a conclusion

The lines 486-502 have been summarized as you requested, in the new version of our manuscript.

Reviewer 2 Report

The purpose of the manuscript is clearly explained and the rationale for the study seems solid. However, I have two major concerns about both the study population and the application of the NPC test.

Study population. The authors consider the same population considered in a previous study: Two hundred subjects (173 females and 27 males; median age, 37 years), 81 (71 females and 10 males) who were diagnosed with HT and the remaining 119 (102 females and 17 males) as controls. A few analyses conducted on this population suggested the purpose of this paper. It seems to me that the reasoning has a kind of circularity that potentially introduces a lot of bias. Indeed, as stated by the authors:

a)  OS has a role in thyroid autoimmunity;

b) in the previous study, HT subjects reported higher intake frequencies of animal foods compared with controls, who reported higher intake frequencies of plant foods and the number of subjects who preferentially consumed poultry instead of red/processed meat was lower in HT subjects than in controls (taken from the abstract of [12]);

c) in the previous study [12], strong evidence on the role of AGEs as a reliable marker of oxidative stress in HT has been provided.

Then, in this manuscript, a relationship among dietary and OS markers is investigated among the same subjects. This “circularity” is somehow recognized by the Authors at lines 444-452. Leaving aside discourses of feasibility, the study would certainly have more depth if it had been conducted on a new population. In my opinion, stratification is not sufficient to solve this issue.

Finally, sex differences in oxidative stress parameters and oxidative damage markers should be carefully discussed, since over 85% of subjects are female.

Application of the NPC test. The Authors stated that they adopted this test as it represents a permutation solution for several multidimensional hypothesis-testing problems in the context of non-parametric inference. Could the authors specify the advantages over non-parametric ANOVA adjusted for multiple testing? The NCP is usually applied to a multivariate response where the variables show some correlation each other (as in imaging). In particular, it is used to test complex hypotheses. In their discussion of the results, the authors consider only the partial tests, and discuss the outcomes of the combined tests only briefly. Applying an NPC test, I would have expected the opposite: instead their argument is typical of the post-hoc analysis of ANOVA. Figure 2 is an instance of this. In summary, I'm not entirely convinced that this method of analysis has brought any benefits in terms of results.

I would like to suggest the Authors to make an effort in discussing the results of combined tests as first point. More in general, the Authors should revise the Discussion: some of the stated results are not entirely supported by their analysis. An example of this is at lines 459-461. It seems to me that significant findings in some (but not all) subgroups are discussed without reference to the specific subgroups.

 Minor remarks

Could the authors specify how an active lifestyle was determined? By a single question? by IPAQ, in some other way? Could they also specify if a validated questionnaire was used to distinguish between (semi)vegetarian subjects and omnivorous?  

Figure 1 would be more interesting if it presented cakes with 4 slices, crossing eating habits with the other factors considered. This bring me to the next comment. It is well-known that confounding arises when a factor is associated with both the exposure (or treatment) and the outcome. So, an analysis of these statistical associations could be useful.

Tables 1-7 report means and standard deviations of variables that have been previously stated being non-Gaussian. Therefore, it would be preferable to report (also) the quartiles.

Minor editing of English language required. For instance , at line 108 "A condition of OS may be the consequence of physiological events (aging and cell 108 senescence)," the term "processes" could better replace "events". At line 275, study population can better replace "casuistry".

Author Response

Comments and Suggestions for Authors

The purpose of the manuscript is clearly explained and the rationale for the study seems solid. However, I have two major concerns about both the study population and the application of the NPC test.

1) Study population. The authors consider the same population considered in a previous study: Two hundred subjects (173 females and 27 males; median age, 37 years), 81 (71 females and 10 males) who were diagnosed with HT and the remaining 119 (102 females and 17 males) as controls. A few analyses conducted on this population suggested the purpose of this paper. It seems to me that the reasoning has a kind of circularity that potentially introduces a lot of bias. Indeed, as stated by the authors:

  1. a)  OS has a role in thyroid autoimmunity;
  2. b) in the previous study, HT subjects reported higher intake frequencies of animal foods compared with controls, who reported higher intake frequencies of plant foods and the number of subjects who preferentially consumed poultry instead of red/processed meat was lower in HT subjects than in controls (taken from the abstract of [12]);
  3. c) in the previous study [12], strong evidence on the role of AGEs as a reliable marker of oxidative stress in HT has been provided.

Then, in this manuscript, a relationship among dietary and OS markers is investigated among the same subjects. This “circularity” is somehow recognized by the Authors at lines 444-452. Leaving aside discourses of feasibility, the study would certainly have more depth if it had been conducted on a new population. In my opinion, stratification is not sufficient to solve this issue.

We thank Referee 2 for this pertinent comment. We agree with the Referee that a further studies are needed, including a new (and larger) population, to analyze more in depth the impact of diet on oxidative stress and related disorders, and our study group is working on this project.  We are aware that of the limitations of re-evaluating  data that has  been already analyzed, but we tried to avoid circulating analysis biased by modifying the classification parameters (the presence of a TH in the previous set, the dietary habits in the present set).  Despite the limitation of analyzing the same study population, the present study was aimed at confirming the role of dietary habits and components in oxidative stress, irrespective of a concomitant autoimmune disorder. In the previous study, we subdivide our population according to the presence/absence of an autoimmune thyroiditis (HT), and compared the serum levels of oxidative stress biomarkers between patients and controls to demonstrate that a condition of oxidative stress occurs in patients suffering from autoimmune thyroiditis even when thyroid function is normal (ie. autoimmunity itself causes OS). Then, we evaluated any influence of dietary habits on the appearance of the disease, reporting the consumption of red meat and a low grade of adherence to the Mediterranean diet were independently associated with thyroid autoimmunity and increased oxidative stress, by means of regression models. Now, in the present study, we changed our perspective and subdivided patients according to dietary habits to evaluate if diet has an impact on oxidative stress regardless of the presence of an autoimmune disease, by means of a different statistical approach, the NPC test (based on the permutation solution), that allowed us to reduce biases related to the relative small sample size and to the prevalence of females in the study population.

2)  Finally, sex differences in oxidative stress parameters and oxidative damage markers should be carefully discussed, since over 85% of subjects are female.

Once again we thank Referee 2 for this very pertinent observation. Most of our study population is represented by females, because thyroid disorders are more common in female than in male subjects and maybe because female subjects are more interested in joining educational campaigns for the general population and/or more motivated to check their thyroid status than males. It is well known that sex differences exist in OS stress mechanisms, since males display a higher production of ROS and less efficient antioxidant mechanisms. There are relevant gender differences in a number of oxidative and inflammatory markers and they may account for the differential lifespan between sexes, as well as for differences in health outcomes between genders.

We have highlighted this point in the discussion and a pertinent reference was added [Martínez de Toda I, González-Sánchez M, Díaz-Del Cerro E, Valera G, Carracedo J, Guerra-Pérez N. Sex differences in markers of oxidation and inflammation. Implications for ageing. Mech Ageing Dev. 2023 Apr;211:111797. doi: 10.1016/j.mad.2023.111797]. 

We have added the following sentence before the conclusion paragraph. “Limitations of the present study were the relatively small number of recruited subjects and rather high prevalence of female subjects in the study group. Indeed, it is well known that sex differences exist in OS stress mechanisms, since males display a higher production of ROS and less efficient antioxidant mechanisms, and the existence of gender differences in oxidative and inflammatory markers should be taken into account [vedi sopra ref da aggiungere]. Consequently, this study group cannot be considered representative of the general population, and further studies on novel and large population are needed to analyze more in depth the impact of diet on oxidative stress and related disorders”.

Application of the NPC test. 

The Authors stated that they adopted this test as it represents a permutation solution for several multidimensional hypothesis-testing problems in the context of non-parametric inference. Could the authors specify the advantages over non-parametric ANOVA adjusted for multiple testing? The NCP is usually applied to a multivariate response where the variables show some correlation each other (as in imaging). In particular, it is used to test complex hypotheses. In their discussion of the results, the authors consider only the partial tests, and discuss the outcomes of the combined tests only briefly. Applying an NPC test, I would have expected the opposite: instead their argument is typical of the post-hoc analysis of ANOVA. Figure 2 is an instance of this. In summary, I'm not entirely convinced that this method of analysis has brought any benefits in terms of results.

The NPC methodology has some advantages over the non-parametric ANOVA adjusted for multiple tests: firstly, it is of the multivariate type, therefore it allows for the simultaneous testing of K-variables of interest (MANOVA). In addition, assuming that the exchangeability of the data between the groups in null hypothesis holds, it is characterized from two important properties:

  • property of similarity, as whatever the distribution underlying the data, the probability of rejection under the null hypothesis is invariant with respect to the set of data actually observed, and this whatever the method of disclosure of the data;
  • for any level of significance α, for any underlying distribution and for all possible observed datasets, if under the alternative the distribution dominates the null or is dominated by the null, then there exists an undistorted conditional test, in the sense that the probability of rejection of the null hypothesis is always lower than the significance level α.

These two properties, respectively, guarantee that the conditional probability of rejecting H0, when it is true, is always equal to the significance level α, regardless of how the data are detected, and that the conditional probability of rejecting H0, when H1 is true, it is always not less than the chosen level of significance α, under the condition, however, that the data can be exchanged between groups. Thanks to these two characteristics, the inferences associated with permutation tests can be extended to the entire target population, respecting the properties of non-distortion and consistency (Pesarin and Salmaso, 2010).

Therefore the advantages of the NPC methodology compared to the non-parametric ANOVA adjusted for multiple tests are attributable to a greater control of the type I error and of the familywise error rate (FWER) in the presence of more variables to analyze. Therefore, NPC offers a link between combining multiple tests and fixing for multiple tests, in both cases regardless of any dependencies between those tests (Pesarin, 2001)

I would like to suggest the Authors to make an effort in discussing the results of combined tests as first point. More in general, the Authors should revise the Discussion: some of the stated results are not entirely supported by their analysis. An example of this is at lines 459-461. It seems to me that significant findings in some (but not all) subgroups are discussed without reference to the specific subgroups.

With reference to the comment of the combined tests, we acknowledge that we have overlooked this aspect. We thank the referee for the useful suggestion that allows us to improve the quality of our article and, in the revised version of the manuscript, we have added specific comments for the combined tests obtained by applying the NPC methodology (lines 380-380 and, also, 513-517 - revised version of the manuscript).

Minor remarks

Could the authors specify how an active lifestyle was determined? By a single question? by IPAQ, in some other way? Could they also specify if a validated questionnaire was used to distinguish between (semi)vegetarian subjects and omnivorous?  

We thank the Referee for this comment. We have now provided more details on data collection as supplementary materials.

Figure 1 would be more interesting if it presented cakes with 4 slices, crossing eating habits with the other factors considered.

Figure 1 was created to globally describe the entire sample of examined subjects, regardless of diet. For this reason it is presented before the identification of groups. In order to accommodate the referees' suggestion, we added another figure (Bar chart, Figure 1b) where each confounding factor is presented distinguishing between omnivores and semi-vegetarians. The additional inclusion of this new figure (Figure 1b) has led to the change in the name of Figure 1 which, in the revised version of the manuscript, becomes “Figure 1a”.

This bring me to the next comment. It is well-known that confounding arises when a factor is associated with both the exposure (or treatment) and the outcome. So, an analysis of these statistical associations could be useful.

We absolutely agree with the referee and thank him for the helpful advice; we had not actually thought to evaluate the associations between diets and confounding factors. Instead, this analysis is certainly appropriate.  In Figure 1b for each bar chart we now report the value of Chi square test and the p-value, referred to the analysis of the statistical association between each confounding factor and the dietary regimes. We have added this analysis also in the text of the manuscript (lines 301-306 of the revised version of manuscript). Also a brief commentary on this association analysis was also inserted immediately after Figure 1b

Tables 1-7 report means and standard deviations of variables that have been previously stated being non-Gaussian. Therefore, it would be preferable to report (also) the quartiles.

The NPC methodology allows, by means of a conditionally Monte Carlo resampling procedure (CMC), to generate permutations of the same data, obtaining a p-value in a non-parametric way, using a two-phase algorithm and an appropriate combination function. “The NPC consists, in a first phase, of testing each hypothesis separately using permutations that are performed synchronously between the datasets (called partial tests). The resulting statistics for each individual permutation are recorded, allowing one to construct a complete empirical null distribution estimate for each one. In a second step, the empirical p-values ​​for each statistic are combined, for each permutation, into a joint statistic. Since such a combined joint statistic is produced from the previous permutations, an estimate of its empirical distribution function is immediately known, as is the p-value of the joint test” (Pesarin, 2001)

We are aware that, in presence of non-parametric tests, descriptive statistics should usually be based on location measures (median and quartiles). But in the applications of this methodology, since the same observations are used and not the transformations into ranks, it is usual to present the variables as mean ± standard deviation (Pesarin and Salmaso, 2010,  p.121, 125, 127). For this reason, Tables 1-7 report the mean ± standard deviation, and the p-values of the comparisons.

Minor editing of English language required. For instance , at line 108 "A condition of OS may be the consequence of physiological events (aging and cell 108 senescence)," the term "processes" could better replace "events". At line 275, study population can better replace "casuistry".

We have replaced the term “events” with “processes” and, also, the term “casuistry” with the expression “sample of subjects”, as you suggested.

Round 2

Reviewer 2 Report

Please, see the attached file for a few further comments.

Author Response

I thank the authors for the detailed response. I kindly suggest that the authors extend the paragraph devoted to the limitations of the study by acknowledging the role of double data analysis.

We certainly accept and appreciate the referee's proposal; we added in the final paragraph of the limitations of the study (before the conclusions) the following sentence:

“Finally, we acknowledge the limitations of a double data analysis since we have re-evaluated data that has been already analyzed by means of a different statistical approach. However, we tried to avoid circulating analysis biased by modifying the classification parameters (the presence of a TH in the previous analysis, the dietary habits in the present one).”

I thank the authors for the detailed answer, which clarifies the general methodological aspects of the adopted method. I remain, however, puzzled by the benefits of this study in particular. I take this opportunity to integrate my previous comments. On lines 310-311 the notation

is used without being explained. That should be the equal distribution symbol, right? In other words, the test verifies whether there are overall differences between the multivariate distributions of the group outcomes.

 The symbol used in lines 310-311 refers to equal distribution symbol, exactly as intuited by the referee. We actually took it for granted, without specifying. We apologize for this.

 In the revised version of the paper we explained the symbol and its meaning, specifying that

“In the hypotheses system, the distributional equality stated by H0 implies that the observed data vectors are exchangeable between two groups. Without loss of generality, we suppose that for each sub-hypothesis H0i against H1i there is a suitable partial permutation test assumed to be significant. The system of hypotheses is set in such a way that the related partial tests are jointly processed, so that they can be nonparametrically combined by taking into account their underlying dependence structure within the nonparametric combination method. Our hypotheses system, that verifies whether there are overall differences between the multivariate distributions of the group outcomes, is the following:…”

After hypothesis system, we specified as followed reported:  “where  refers to equal distribution symbol” and, at the end of the sentences, we added the sentence: “Global H0i  hypothesis is true if all sub-null are jointly true (for intersection criterion), the alternative hypothesis H0i is true when at least one sub-alternative is true (for union criterion)”.

This also leads me to reiterate that reporting only means and variances in Table 1 is not appropriate for the context, and to suggest once again adding quartiles.

In accordance with the referee's recommendation, in the new version of the manuscript we added median (and in brackets the 1st and 3rd quartile) in tables 1-7.

Thanks to the authors for adding the previous comments. However, I believe the picture is still incomplete. For example, can we reject the hypothesis that the multivariate distribution of parameters in males is equal to that in females? If yes, at what level of significance? The results presented do not answer this question, it seems to me.

In our hypothesis system we do not test that the multivariate distribution of parameters in males is equal to that in females; the grouping variables is dietary regimes (omnivorous vs semi-vegetarian subject). We verify, through stratification, if such differences between two dietary regimes exist in some strata of subjects, defined by the confounders (gender, age class etc…).

In reference to significance level, we specified in line 247 of manuscript that “For all statistical analyses, the significance a level was set at 0.05”.

 Moreover, in order to better clarify this aspect, we inserted in the manuscript (section Material and methods”) these following sentences:

The NPC methodology has some advantages over the non-parametric ANOVA adjusted for multiple tests: firstly, it is of the multivariate type, therefore it allows for the simultaneous testing of K-variables of interest (MANOVA). In addition, assuming that the exchangeability of the data between the groups in null hypothesis holds, it is characterized from two important properties:

  • property of similarity, as whatever the distribution underlying the data, the probability of rejection under the null hypothesis is invariant with respect to the set of data actually observed, and this whatever the method of disclosure of the data;
  • for any level of significance α, for any underlying distribution and for all possible observed datasets, if under the alternative the distribution dominates the null or is dominated by the null, then there exists an undistorted conditional test, in the sense that the probability of rejection of the null hypothesis is always lower than the significance level α.

These two properties, respectively, guarantee that the conditional probability of rejecting H0, when it is true, is always equal to the significance level α, regardless of how the data are detected, and that the conditional probability of rejecting H0, when H1 is true, it is always not less than the chosen level of significance α, under the condition, however, that the data can be exchanged between groups. Thanks to these two characteristics, the inferences associated with permutation tests can be extended to the entire target population, respecting the properties of non-distortion and consistency (Pesarin and Salmaso, 2010).”

The argument produced by the authors is not entirely correct. Presenting quartiles, as well as mean and standard deviation, serves to better describe the distribution of data, which is exactly what permutation methods are supposed to define and use. It is not, however, dependent on using data ranks in tests.

In accordance with the referee's suggestion, in the revised version of our manuscript we presented the data in Tables 1-7 not only as mean ad standard deviation, but adding quartiles (specifically median, Q1 and Q3) in order to better describe the distribution of the data, which is exactly what permutation methods define and use. We have added in the manuscript this last sentence “in order to better describe the distribution of the data, which is exactly what permutation methods define and use”, for which we thank the referee.
